

# A first wildfire risk assessment for Belgium

Arthur Depicker[1], Bernard De Baets[1], and Jan M. Baetens[1]

[1]KERMIT, Dept. of Data Analysis and Mathematical Modelling , Ghent University , Coupure links 653, 9000 Ghent, Belgium

**Correspondence:** Jan M. Baetens (jan.baetens@ugent.be)

**Abstract.** This paper presents a wildfire ignition probability map for Belgium that may serve as a proxy for wildfire risk. Wildfires have proven to cause considerable damage to valuable Natura 2000 areas in Belgium in the past, and the prevalence of such events is anticipated to increase in the future. Therefore, a risk map can be used to optimize the allocation of wildfire prevention and fighting resources. Even more, such a map is required by EU legislation in order to receive financial support for wildfire-related forest and nature management. Firstly, clear definitions of 'wildfire risk' and 'wildfire hazard' are given to avoid misconceptions of the used wildfire terminology. The former is assessed as the 'wildfire ignition probability', while the latter is regarded as the 'fire potential' and is assumed to be solely determined by the characteristics of the vegetation. Secondly, an actual wildfire hazard map is designed using an existing expert system. Thirdly, a wildfire ignition probability map is constructed by relying on Bayes' rule and using spatial wildfire ignition data, on the one hand, and spatial data on land use, soil type, and land cover, on the other hand. It appears that the most wildfire prone areas in Belgium are located in heathland where military exercises are held. The provinces that have the largest relative areas with a high or very high wildfire risk are Limburg (24.67%), Antwerp (20.64%), and Luxembourg (4.54%). Our study also reveals that most wildfire ignitions in Belgium are due to humans (arsonism, cigarettes,...), and that natural causes such as lightning are rather scarce.

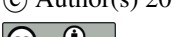



## Introduction

Every year, wildfires globally burn an astonishing 350-450 million hectares of forest and grassland, an area corresponding to approximately 4% of earth's land surface, Antarctica and Greenland not taken into account (Randerson et al., 2012; Giglio et al., 2010). The general perception is that wildfire frequency and damage are increasing (National Wildlife Federation, 2008; North et al., 2015; Doerr and Santin, 2016) due to more extreme weather events and altered precipitation and temperature patterns (IPCC, 2014). Predictions of wildfire prevalence considering these future climatic conditions show an increase in some regions, and a decrease in others. Hence, global change is likely to redistribute wildfire prevalence, rather than simply increasing it homogeneously across the world (Krawchuk et al., 2009). In Europe, predictions are particularly ominous for southern Europe, where an increasing prevalence of droughts, heat waves, and dry spells are expected to elevate the future wildfire size and intensity (Moreno, 2014). Likewise, such events are likely to increase the occurrence of wildfires in temperate regions, such as Belgium and Germany (European Environment Agency, 2015), which do not have a rich wildfire history.

## Wildfires in Belgium

### Prevalence

The prevalence of wildfires in Belgium is rather limited. The annual burnt area rarely exceeds 40 hectares, but depending on the meteorological conditions relatively large areas –in a Belgian context– can be affected. Unfortunately, these fires often occur in biologically valuable nature areas. In 2011, for instance, 2144 hectares of land burnt within the Natura 2000 network, consisting of protected nature areas throughout the European Union (Miguel-Ayanz et al., 2012). In this paper, a more detailed assessment of wildfire prevalence in Belgium is executed. The results are presented in Section .

### Damage

Since even the vaster wildfires in Belgium did not damage infrastructure or housing, while there have been no human casualties up to this day, it may be concluded that the damage cost of wildfires in Belgium is very limited. Essentially, wildfire damage occurs most frequently in natural areas, where wildfires might jeopardize the survival of vulnerable species like *Lyrurus tetrix* (Jacob and Paquet, 2011) or promote the growth of undesired plant species such as competitive grasses (e.g. *Molinia cearulea*) that suppress the presence of characteristic plant species, such as *Calluna vulgaris* and *Erica tetralix* (Marrs et al., 2004; Jacquemyn et al., 2005; Schepers et al., 2014). Hence, wildfire research in Belgium is important from a biological, ecological, and nature conservation perspective.

In that respect, it is important to estimate the monetary value of nature in Belgium. Focusing on Flanders, Liekens et al. (2013) did this on the basis of a large-scale choice experiment to determine the willingness of households to pay for nature (€/household/year). These authors rank forest as the most valuable (€182), followed by heathland and inland dunes (€159), grassland (€158), open reed and swamp (€146), pioneer vegetation (€119), and marshes (€117). These monetary values should not be used to determine the value of nature areas, but rather to compare the value of different types of nature. It should



also be noted that the monetary value of a burnt nature area is not necessarily affected in the long run since regeneration of the vegetation will often occur. Still, wildfires can alter the monetary value of an area if its cover changes from one type of nature to another. Even so, monetary value does not necessarily reflect ecological value.

**Detection and Suppression**

As a consequence of the high population density, wildfires in Belgium are rapidly detected and reported to the emergency services. Moreover, in some valuable nature areas extra efforts are made for an even more rapid detection. For instance, on days with a (very) high wildfire risk, in one of Flanders vastest nature areas (Kalmthoutse Heide), volunteers man a fire watch tower - a building structure that offers a clear view of the area, and immediately report any detected smoke or flames to the emergency services. Currently, this is the only way of wildfire detection in use.

Wildfires are suppressed by ordinary firefighters using their standard equipment, which is complemented with dedicated terrain vehicles to gain access to rough terrain, while some firefighters got a specific training in France (Federal Public Service Internal Affairs, 2013). Belgium also lacks planes or helicopters that can be deployed in the case of wildfires, though in 2015 a bilateral agreement between Belgium and The Netherlands was signed to deploy a dedicated helicopter from The Netherlands in the case of major events (Ministry of Justice and Security, 2015). Also in the past, aerial means from neighboring countries

were deployed in large-scale exercises in the Hoge Venen to fight the largest wildfires (Belga, 2013). Since wildfires are rather rare and mostly ordinary firefighting equipment is used, the suppression cost of wildfires in Belgium is expected to be a limited portion of the total budget spent to its emergency services.

**Prevention**

The main prevention strategy in nature areas is to assign a color code reflecting the wildfire risk that is assessed by the terrain

manager and local experts, and comes with specific guidelines for visitors and firefighters depending on the color code. 'Code green' means that there is a low wildfire risk, and in the unlikely event of a wildfire, the fire brigade follows the standard procedure in terms of the number of men. 'Code yellow' is associated with an elevated risk. For instance, in the Kalmthoutse Heide the watch tower is manned on such days. If a wildfire is detected in a region with 'code orange', the fire brigade will deploy extra men and equipment. Moreover, the fire watch tower is permanently manned and children can only play under

parental supervision. Finally, 'Code red' means that the wildfire risk is very high and access to such areas is discouraged (ANB, 2017).

Another form of prevention is the construction or repair of firebreaks, as illustrated in the management plans for the military domains of Helchteren and Vrijbos-Houthulst (Vandenberghe et al., 2009; Waumans et al., 2009).

**National Action Plan Wildfires**

In the aftermath of the 2011 wildfires (Miguel-Ayanz et al., 2012), and largely motivated by the shortcomings and problems detected while being faced when fighting relatively vast wildfires (up to 1000 ha), the National Action Plan Wildfires was



compiled by the Directorate-General of the Federal Public Service Internal Affairs in order to evaluate and improve the risk analysis and cartography, materials, procedures and training, emergency planning, and exercises related to the outbreak of wildfires (Federal Public Service Internal Affairs, 2013). Although a preliminary risk map was constructed based on the qualitative feedback from emergency planning services and province governors, EU legislation dictates that a more scientifically

sound approach should be used. This is important because the law states that forest areas classified as medium to high forest fire risk are eligible for financial support of the European Regional Development Fund. However, such a wildfire risk map must be supported by scientific evidence and acknowledged by scientific public organizations, in agreement with Article 24 of Regulation (EU) No 1305/2013 of the European Parliament and of the Council of 17 December 2013 (the European Parliament and the European Counsil, 2013). In order to support the EU member states in arriving at such a map and to harmonize the

used methodology across the EU member states, the European Commission has consulted the EU member states on how the JRC should proceed during the 2017 meeting of the Commission Expert Group on Forest Fires (REF). Moreover, the preliminary risk map included in the National Action Plan Wildfires did not account for how high risk is perceived differently by the consulted parties across the country.

**Wildfire Hazard and Risk**

The term 'wildfire hazard' refers to the potential fire behavior, which is usually assessed through the so-called fuel complex. This complex relates to the vegetation and its physical characteristics, such as volume and moisture content, and determines the ease of ignition and of its resistance to control (National Wildfire Coordinating group, 2012; Hardy, 2005).

Since the term 'fire hazard' only expresses the potential for wildfire occurrence, there is still a need for causative factors (i.e. sources of ignition) to convert hazard into risk (Hardy, 2005; Srivastava et al., 2014). Examples of such factors are lightning,

smoking, campfires, military exercises, and arsons.

In wildfire literature, the term 'risk' is generally used in two different ways. On the one hand, it can refer to the possibility of loss, harm and injury. On the other hand, it can simply refer to the probability of occurrence of an event (Bachman and Allgöwer, 2000). The 'fire risk' definition that is commonly accepted by the fire community is the following: 'The chance that a fire might start, as affected by the nature and incidence of causative agents (Hardy, 2005).' In this paper, the latter definition

will be used. More specifically, in the remainder of this manuscript, the term 'wildfire risk' refers to the probability that a wildfire ignition might occur.

A (spatial) wildfire risk assessment can provide information to support decision-making, for example, to locate fire-fighting resources and to reduce the impact of wildfire events (Thompson et al., 2016; Haas et al., 2013; Salehi et al., 2016). For that reason, the Belgian National Action Plan Wildfires explicitly considers wildfire risk assessment (Federal Public Service

Internal Affairs, 2013).

The risk assessment can be static or dynamic. The former is governed by static variables that can be linked to wildfire hazard (e.g. topography, land use, climate, and average fuel condition), sources of ignition, and data on prior wildfires in the region at stake. This type of assessment provides information for long-term decisions like land use planning and the location of resources. The latter risk assessment is based on real-time information, such as meteorological forecasts and the moisture



condition of vegetation. The goal of such an assessment is to identify the areas with the highest wildfire risk in order to send out patrols and take preemptive action, like alerting local authorities and prohibiting agricultural practices like stub burning. Thus, a dynamic risk assessment is used in short-term decision-making (Fiorucci et al., 2010).

Common approaches for a static wildfire risk assessment involve logistic regression (Martinez et al., 2008; Catry et al.,
2009; Vilar del Hoyo et al., 2011; Preisler et al., 2004) and machine learning techniques (Massada et al., 2012; Rodrigues and de la Riva, 2014). Moreover, a Bayesian weights-of-evidence modeling approach has been used to quantify the ignition risk (Kolden and Weigel, 2007; Dickson et al., 2006). This method comprises the use of Bayes' rule to calculate weights for the different classes of input maps. These weights are then integrated per grid cell in a logit equation to obtain a probablity with value between 0 and 1 (Dickson et al., 2006).

The aim of this paper is to conduct a static wildfire risk assessment for Belgium. In the next section, the materials and methods are discussed. First, the study area is presented together with the spatial data, necessary for the assessment. Second, the applied methods for constructing a hazard and a risk map are exposed. In Section 3, the resulting risk map is presented and compared to the actual wildfire hazard. If the ignition sources are distributed homogeneously over the study area, we expect a strong correlation between hazard and risk. Section 4 provides a discussion on the results, including some recommendations
for future wildfire management in Belgium.

## Materials & Methods

### Study Area: Belgium

Belgium is a western European country and a member state of the European Union. It is bordered by France to the south, Luxembourg and Germany to the east, the Netherlands to the north, and the North Sea to the west. Belgium has a temperate
maritime climate that is characterized by four distinct seasons: spring, summer, fall and winter. It has a total area of approximately 30,528 $km^2$ and a population of more than 11.2 million. The average population density is 363 inhabitants per $km^2$, though the northern region, Flanders, is much more densely populated than the southern region, Wallonia (562 inh./$km^2$ versus 214 inh./$km^2$) (Belgian Federal Government, 2016). The average population density in Flanders is also much higher than in the Mediterranean countries: Spain (93 inh./$km^2$), Portugal (115 inh./$km^2$), France (118 inh./$km^2$), Greece (84 inh./$km^2$),
and Italy (203 inh./$km^2$) (United Nations, 2015). Contrary to these countries, Belgium has no remote areas with low population densities that are not urbanized in one way or another. This implies that wildfires can pose a threat for urban areas as a consequence of the relative proximity of residential or commercial areas (e.g. Kalmthoutse Heide)

### Data

In order to conduct a static risk assessment for the Belgian territory, data on historical wildfires and spatial data such as land
cover, soil, and land use are needed.



Data on wildfire ignitions during the period 1985–2016 were collected in two ways. Firstly, a list of all wildfire interventions between 2010 and 2013 was provided by the Directorate-General of the Federal Public Service Internal Affairs. Secondly, the digital archives of several newspapers were searched. These archives covered the period 1985–2016, though, relevant data was retrieved for the period 1994–2016 only. The following newspapers were searched: *Gazet van Antwerpen*, *Het Laatste Nieuws*,

*Het Belang van Limburg*, *Le Soir*, *L'Echo*, *La Dernière Heure*, *La Meuse*, *La Nouvelle Gazet*, *Metro*, and *L'Avenir*, thereby ensuring that most news items on wildfires throughout the country would be retrieved.

In order to analyze these wildfire data and conduct a wildfire risk assessment, several GIS layers are required. Here, the considered GIS layers are land cover, soil, and land use. Since the study area has an overall high population density, this layer is not included in the analysis. Besides, human activity is more or less integrated in the variable 'land use'. The land cover

vector data set, dating from 2011, was obtained from the Belgian National Geographic Institute (NGI) and rasterized (10m x 10m resolution). The soil and land use vector data were provided by the Flemish Soil Database (DOV) and the Walloon Public Service (SPW) and also rasterized (10m x 10m resolution). The Flemish soil and land use layer date from 2016 and 2014, respectively, while those from Wallonia date from 2007 and 2016, respectively. The data for both the Flemish and Walloon region were merged into one layer, after homogenizing the map legends by assigning similar categories in both layers to a new

class. The resulting layers are displayed in Figure 1. The soil and land cover layer can serve as a proxy to fire susceptibility, as soil texture is correlated with soil moisture and land cover with vegetation. Land use provides insight in human activity, which can be related to ignition causes.

Given the nature of the applied methodology for wildfire risk assessment and the extent of the historical wildfire dataset (See Section ), the number of spatial layers has to be restricted. Spatial data such as precipitation, distance to roads, slope,

aspect, and elevation (Dickson et al., 2006) were excluded from our analysis. Besides, spatial rainfall variability is relatively limited given the extent of the country, and the road network is very dense across the entire country. In fact, the road density in Belgium is five times as high as the average for the European Union (5.1 km/km$^2$ versus 1.1 km/km$^2$) (European Union Road Federation, 2016). Furthermore, in most cases the location of wildfire interventions by firefighters is identified by means of a residential address (municipality, street name and number), possibly biasing the perception of wildfire occurrence in function

of the distance to roads.

## Methods

As discussed in Section , hazard is referred to as the potential fire behavior and is solely determined by the characteristics of the vegetation (National Wildfire Coordinating group, 2012; Hardy, 2005). Hence, in a first step, a wildfire hazard map was derived from the land cover data of the study area. A score between 0 and 100 was assigned to each land cover type, according

to an expert system introduced for the Netherlands (Table 1) (Verboom et al., 2013).

The ultimate goal of this paper is to construct a wildfire ignition probability map on the basis of three categorical variables: land cover class, soil type, and land use class. Hence, in a first step, the non-parametric $\chi^2$ test of independence was used to determine the significance of each of these three categorical variables. In other words, the test determines whether a predictor can be used to explain the variance in the observed and expected relative wildfire ignition frequency (McDonald, 2014). The





expected wildfire ignition frequency for a certain class of one of the variables was calculated as the area of that class divided by the total area of Belgium. The level of significance was set to 5%.

The wildfire risk is assessed in terms of an Ignition Probability Map (IPM) that reflects the probability that a wildfire takes place during the course of one year, on a 10m x 10m grid cell given its features. In total, three wildfire IPMs were constructed.

The first is solely based on land cover class, the second one on land cover class and soil type, and the third one on land cover class, soil type, and land use class.

The ignition probability was calculated by means of Bayes' rule (Dawid et al., 2005):

$$P(I|C_i) = \frac{P(I)\,P(C_i|I)}{P(C_i)},\qquad(1)$$

where $I$ indicates an ignition event and $C_i$ contains the features that characterize the environment of cell $i$, which are inferred

from the GIS layers (Section ). Hence, an environment is a specific combination of features, derived from land cover class, soil type, land use class, or a combination of the aforementioned. Note the difference between the straightforward application of Bayes' rule (demonstrated in this paper) and the Bayesian weights-of-evidence approach (Dickson et al., 2006; Kolden and Weigel, 2007). The latter does not directly use Bayes' rule to calculate the wildfire probability in a grid cell. Bayes' rule is applied to calculate the weight of each input layer in the specific grid cell. It then uses the logit equation to convert these

weights into a probability (with value between 0 and 1).

Because the number of data points is limited (cfr. Section ), the GIS layers were reclassified into fewer classes, in order to reduce the possible number of environments with a distinct set of land cover class, soil type, and land use class. Otherwise, too many environments without any recorded wildfires would be created. In this paper, the maximum number of environments is arbitrarily set at 20.

The first IPM was constructed by taking into account land cover classes, which gave us 10 possible environments. These are displayed in Figure 1 (a).

For the construction of the second IPM, making use of land cover and soil data, land cover was reclassified into three classes on the basis of the discrepancy between the observed and expected number of wildfires as demonstrated by Figure 2 (a): forests (25.44%), by merging deciduous, mixed and coniferous forests, shrubland (2.84%), by grouping heathland and shrubland, and a

third class containing the remaining land cover classes (71.72%). So for the second IPM, 18 environments remained. Likewise, the soil map (Figure 2 (b)) was reclassified into sand (21.35%), wetlands/fens (0.48%), and other soil types (78.17%). The third IPM was based on the simplified land cover and soil maps, and a land use map (Figure 2 (c)) distinguishing between three classes: military domains (1.18%), nature areas (25.43%), and other uses (73.39%). In total, 27 environments were defined in the third IPM. However, this number was reduced to 20 by merging some of the environments for which no or very few

wildfires were registered. For example, all environments that were military domain and not situated on a sandy soil were merged since all of these together contained only one registered wildfire.



In Eq. (1), the probability that a randomly selected cell belongs to class $C_i$ is equal to

$$P(C_i) = \frac{\text{Area of } C_i}{\text{Total Area}}. \tag{2}$$

$P(Ci|I)$ is the probability that, given that an ignition took place in cell $i$, this cell belongs to class $C_i$, and was computed as:

$$P(C_i|I) = \frac{\text{Number of ignitions in } C_i}{\text{Total number of ignitions}}, \tag{3}$$

with the total number of ignitions determined by the number of ignitions used for the construction of the IPM. Finally, the probability that an ignition occurs in a random cell within the time span of one year was calculated as

$$P(I) = \frac{\text{Average annual number ignitions}}{\text{Total number of cells}}. \tag{4}$$

In order to compare the quality of these three different IPMs, each IPM was constructed 23-fold, each time leaving out the wildfire data of one year. The average ignition probability at the wildfire locations of the discarded year was then predicted
by means of the IPM. For example, for the first of the 23 risk maps we used the data between 1994 and 2015 to construct the risk map, and the data of 2016 to check if this risk map predicts a high wildfire ignition risk at those locations where wildfires occurred in 2016. In this way, an indication is obtained of how reliable the map reflects the wildfire risk at locations that were effectively affected in the course of history.

To identify the IPM with the highest average ignition probability for the data points, not used for its construction, we relied
on the non-parametric Mann-Whitney $U$ test (McDonald, 2014), with a 5% level of significance. The IPM resulting in the highest average predicted ignition probabilities in observed wildfire locations was considered to be the most accurate.

Next, the best IPM was further investigated. In order to assess the influence of the number of data points on the quality of this map, we constructed the IPM several times with data sets of varying size. The first map was constructed with data from the period 1994–2000. Subsequently, we incrementally increased the length of the period from which data were used in the
IPM construction stage with one year. As such, we constructed 13 IPMs, the first one with data from the period 1994–2000, the last one with data from the period 1994–2012. The quality of each of these 13 IPMs was assessed by calculating the average ignition probability retrieved at wildfire locations during the period 2013–2016. The robustness of each of the 13 IPMs was tested by constructing each of the risk maps 100 times with 90% of the data, randomly selected. This approach allowed us to construct a boxplot of the corresponding average ignition probabilities in the 13 IPMs. The range of these probabilities is a
proxy for the robustness.

Finally, the best IPM was compared to the hazard map. It is expected that lower hazards will correspond to lower ignition probabilities, and vice versa. So plotting the average ignition probability versus the hazard should result in an increasing curve if ignition sources are distributed homogeneously over the study area.



## Results

### Wildfire Ignition Data

### Spatial Analysis

In total, 385 wildfires were recorded, from which 273 were assigned GPS coordinates. The wildfire locations are displayed

in Figure 3. In Flanders, the northern half of Belgium, the eastern provinces of Antwerp and Limburg clearly show a higher wildfire risk and prevalence than the other provinces. In Wallonia, the southern part of Belgium, wildfires seem to be less rampant and occur mainly in the east and south-west parts of the region. An explanation for the distribution of these wildfires can be sought in the social, economical and technological shifts of the 19[th] century, like the industrial revolution and agricultural innovations (Buis, 1985)

In Flanders, the omnipresent heathland, characterized by poor, sandy soils, was afforested in the eastern provinces with *Pinus sylvestris*, while the forests on the rich soils in the west were cleared for agricultural practices (den Ouden et al., 2010). Present-day, both forests and heathland are relatively more common in Limburg and Antwerp than in the rest of Flanders (Hermy et al., 2004), thus it is expected that the average wildfire risk in these two provinces is higher than in the other Flemish provinces.

In Wallonia, the relative forested area is three times as high as the one in Flanders, 32% versus 11.4% (Walloon Government and the European Commission, 2015; Stevens et al., 2015). The forested areas are mainly concentrated in the eastern provinces of Liège and Luxembourg. The typical tree species that is used for afforestation in this region is *Picea abies*, a coniferous species associated with a very high wildfire sensitivity (Goldammer and Furyaev, 2013), which would explain a relatively high risk in the latter two provinces. As expected, the nature reserve les Hautes Fagnes (in the eastern part of Liège) and its

surrounding area show a higher prevalence because of its fens, which get dry easily in the absence of rain.

Unfortunately, precise data on the size of wildfires were very scarce. Most wildfires covered small areas (<1ha). Though for some major events, relatively accurate estimates of the burnt area could be provided (Miguel-Ayanz et al., 2012). It may be interesting to note that most major wildfires occurred in heathland or fen. It seems that wildfires in such land cover are less controllable than those in coniferous or deciduous forests. This can be understood by the fact that heathlands and fens are

largely covered with shrubs and grass that ignite easily, and hence allow the wildfire to propagate rapidly, once flames have evolved. In 2011, a series of wildfires raged through three nature areas: the High Fens, de Kalmthoutse Heide (heathland) and the military domain in Meeuwen, destroying respectively 1000, 500 and 360 ha. In total, 2180.39 ha of land were burnt that year, mainly NATURA 2000 sites (Schmuck et al., 2012).

### Temporal analysis

Contrary to the statement of the Federal Public Service Internal Affairs in its National Action Plan Wildfires that there are two periods with an elevated wildfire occurrence (i.e. April–May and August) (Federal Public Service Internal Affairs, 2013), the





data displayed in Figure 4 indicate that the wildfire risk is highest in April, then drops rapidly in May and June, and remains stable afterwards until August, after which the relative frequency drops to nearly zero.

Figure 5 shows the number of wildfires per year for the period 1995–2015. The data for 1994 was omitted because almost no newspapers were digitized for this period, and the wildfires for 2016 were not included in the graphic because, at the time this
research was conducted, the year had not yet passed. The figure shows clearly that there is a great variability in the number of wildfires between different years. A critical note is that for the period 2010–2013, the data was more complete (because a list with wildfire interventions was provided by the government), possibly explaining the higher number of wildfires in these years. Still, in 2003, the number of wildfires is extremely high, as a consequence of the extremely warm and dry summer (Eysker et al., 2005).

**Ignition Sources**

This research made it clear that cigarettes, arson and military exercises were major drivers of ignition, even records have been found that support the hypothesis that pieces of glass can trigger a fire through the redirection and focusing of sunlight (Timperman and Willekens, 1999). No reports were found of natural ignition causes such as lightning. In other words, humans are the main driver of wildfires in Belgium. This is consistent with other regions in Europe, e.g. the Mediterranean area where
95% of the ignitions can be attributed to human causes (San-Miguel-Ayanz et al., 2012)

**Hazard Map**

The wildfire hazard map (Figure 6) was constructed by scoring each land cover class in agreement with the scores listed in Table 1. According to this map, the most hazardous regions in Belgium are located in the Eastern provinces, namely Antwerp, Limburg, Liège, and Luxembourg. The former two are situated in the northern region of Belgium, Flanders. The latter two
belong to the Southern region, Wallonia. This can be explained by the relatively large areas of forests and natural vegetation in these provinces, on the one hand, and the presence of coniferous forests, which are prone to wildfires, on the other hand (Wijdeven et al., 2006).

**Ignition Probability Map**

Figure 2 shows the comparison between the observed and expected ignition frequencies for each variable, whereby the expected
ignition frequency is calculated a the proportion of the total study area of each category of that specific variable. As the non-parametric $\chi^2$ test of independence proved, the land cover class clearly influenced the wildfire ignition probability ($\chi^2 = 206.4$, $p < 0.05$). Likewise, soil type had a significant impact on the prevalence of wildfire ignitions ($\chi^2 = 100.4$, $p < 0.05$), as did land use class ($\chi^2 = 198.2$, $p < 0.05$).

Figure 7 shows, for each of the three IPMs, the 23 different average wildfire ignition probabilities, observed at the wildfire
locations that were not used for the IPM construction. The Mann-Whitney $U$ test learns that there is no significant difference in the medians of the IPM based on land cover class, and the IPM based on land cover class and soil type ($p = 0.7584$). However,





the IPM based on land cover class, soil type, and land use class has a significantly higher median than the IPM based on land cover class ($p < 0.05$) and the IPM based on land cover class and soil type ($p < 0.05$). The IPM based on land cover class, soil type, and land use class is therefore considered to provide the best indication of wildfire risk.

From Figure 8, we infer that the quality of the IPM increases with a growing number of data points used for its construction, though this effect only manifests itself clearly for data sets with data from 1994 to 2010 or later. It can also be observed that the robustness of the IPM increases substantially for the smaller data sets, while, for datasets larger than the one that contains the data from the period 1994–2004, the quantiles of the boxplots appear at more or less the same values.

The final IPMs were constructed with all 273 data points. Table 2 shows the area of each risk class in the three different IPMs. These classes were defined on the basis of visible gaps in the histograms showing the relative frequency of the number of registered wildfires located in an environment with a particular ignition probability (Figure 9). The class limits were selected in such a way they were equal for each risk map. The ignition risk classes were 'low', 'intermediate', 'high', and 'very high', corresponding to the ignition probability intervals $[0, 0.025\times10^{-6}]$, $[0.025\times10^{-6}, 0.12\times10^{-6}]$, $[0.12\times10^{-6}, 0.4\times10^{-6}]$, and $[0.4\times10^{-6}, 1]$, respectively.

The IPM leading to the highest probabilities assigned to the wildfire ignition points is the one that considers land cover class, soil type, and land use class; hence, such an IPM was constructed with all 273 ignition points (Figure 10). The average ignition probability assigned to all data points was $0.045 \times 10^{-6}$ wildfire ignitions per year and per 10m x 10m grid cell. The relative area per risk class for each province is presented in Table 3.As expected for Flanders, the provinces of Antwerp and Limburg have the largest high-risk area. In Wallonia, the provinces of Liège and Luxembourgh appear to be most sensitive to wildifres.

The maximum calculated probability for the final IPM was $0.85\times10^{-6}$, so this means that at most one may expect such an area of, for instance 100 ha, to be affected by a wildfire once every 118 years. This value is the expected value, associated with the probability that at least one of the grid cells within the area will burn in the time span of one year, calculated with following equation:

$$P_A = 1 - (1-p)^{N_A}, \tag{5}$$

where $P_A$ is the probability that a certain area $A$ will be affected by a wildfire in the span of one year, $p$ is the probability that a grid cell will burn within one year (as calculated with Equation 1), and $N_A$ is the number of grid cells within area $A$.

The maximum calibrated probability is extremely low compared to the results obtained with logistic regression or machine learning techniques. Catry et al. (2009), for example, obtained a probability higher than 0.8 for 2.6% of the Portuguese land surface, using logistic regression. Similarly, Martinez et al. (2008) and Massada et al. (2012), respectively using logistic regression and machine learning techniques, obtained a value higher than 0.8 for a significant portion of the study area. Yet, these values should not be interpreted as ignition probabilities, but rather as a similarity measure between the spatial characteristics of a given pixel versus the average spatial characteristics of historical wildfires.

The main limitation of approaches leaning on logistic regression, machine learning techniques, and Bayesian weight-of-evidence, is the lack of time-specificity. If the ignition probability in a grid cell equals 0.8, how should this be interpreted?



Is there a probability of 80% that this pixel will burn in a given year? Using a straightforward application of Bayes' rule, the calculated probability does not lack time-specificity. The calculated values in Figure 10 reflect the probability that a grid cell (10m×10m or 100 m$^2$) will contain a wildfire ignition within the period of one year.

**Hazard Map versus Ignition Probability Map**

Figure 11 shows the average ignition probability versus the wildfire hazard. It demonstrates that the probability does not increase with growing hazard, but is significantly lower for the highest hazard score than for a hazard score of 60. On the one hand, this is a consequence of reclassifying the land cover map in such a way that it does not discriminate between deciduous forest (hazard score 20) and coniferous forest (hazard score 100). On the other hand, the IPM solely based on land cover class also shows that the ignition probability in 'mixed heathland and forest' (hazard score 60) is much higher than in coniferous forest, i.e. $1.29 \times 10^{-6}$ versus $0.15 \times 10^{-6}$ wildfire ignitions per year and per 10m x 10m grid cell. Therefore, the

low probability at hazard score 100 cannot be fully explained by the merging of deciduous and coniferous forest into one land cover class. Besides, the ignition probability in coniferous forest is less than two times higher than in deciduous forest, which corresponds to a ignition probability of $0.09 \times 10^{-6}$ wildfire ignitions per year per 10m x 10m grid cell. Military exercises appear to be one of the major ignition causes, thereby also explaining the discrepancy between wildfire hazard and risk. Most

military exercises are held in heathland. This land cover class is less hazardous than coniferous forest (Table 1), yet it has a higher risk due to the exercises

**Discussion**

It should be underlined that this is a very first assessment of the wildfire risk in Belgium, which was complicated by two important factors. Firstly, literature on wildfires in Belgium is limited, and existing papers are often restricted to a description

of the effects of wildfires on ecosystems (Marrs et al., 2004; Jacquemyn et al., 2005; Schepers et al., 2014). Secondly, the considered dataset was very small, not only because wildfire are relatively rare in Belgium, but also because they were not always registered (as such) by the intervened emergency services. Only 273 wildfires were recorded over the last 20 years. As a consequence of the limited dataset and the applied method, only few spatial data layers can be used for the wildfire risk assessment. However, due to an increased interest of policy makers in wildfires (Federal Public Service Internal Affairs, 2013),

partly motivated by the fact that wildfires might occur more frequently in the future, and a standardized registration of fire brigade interventions, it can be expected that more data will become available in the near future.

In this paper, wildfire risk was defined as the probability that a wildfire will occur (Hardy, 2005). The value of damage caused by wildfires to properties and human livelihoods was not included in the analysis due to the fact that no such records could be found. Considerable damage only occurred in nature areas, for instance like in 2011, when 2144 hectares of natural

areas were consumed by flames within the Natura 2000 network.

As to the allocation of wildfire prevention and control resources, Table 3 could serve as a guideline. Provinces with a higher wildfire risk should receive a proportionally higher share of the available resources, and dedicated vehicles and equipment



for fighting wildfires should be stationed near the most vulnerable areas. Besides, the IPM can be used to identify extremely fire-prone areas, where, for instance, the construction of watch towers could be considered.

A final remark is that most causative factors are human. Anthropogenic ignition causes such as arson, cigarettes, campfires, and waste glass have been reported, while natural causes such as lightning appear to be exceptional. All together, it seems

that the best way of preventing wildfires is perhaps to exclude human activities in the most fire prone areas and increase the awareness among the general public, so that people become more aware of the danger they pose to their natural environment.

# 1   Conclusions

Temperate regions like Belgium are expected to face wildfire more frequently in the future as a consequence of climate change. For that reason, a very first wildfire risk assessment for Belgium was performed as a first step in getting the country prepared

for adaptation and mitigation to climate change. Using a probabilistic framework, we found that the most wildfire-prone areas in Belgium are located in heathland and that the provinces of concern are Limburg, Antwerp, and Luxembourg.

*Data availability.* The used land cover data set can be requested from the Belgian National Geographic Institute, while the used land use and soil data can be obtained from the Flemish Soil Database (DOV) and the Walloon Public Service (SPW). Moreover, both the historical wildfire data and the GIS layer relating to the resulting wildfire risk map are available upon request from the authors.

*Author contributions.* Arthur Depicker was responsible for the data processing and analysis, while the three authors took part in compiling the manuscript.

*Competing interests.* No competing interests are present.

*Acknowledgements.* We would like to thank the Belgian National Geographic Institute for providing the land cover data set, while we are grateful for the land use and soil data provided by the Flemish Soil Database (DOV) and the Walloon Public Service (SPW).





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




**Tables**

**Table 1.** Wildfire hazard according to land cover type Verboom et al. (2013).

| Category | Land cover | Score |
|---|---|---|
| Not inflammable | Water, pasture, farmland | 0 |
| | Land dunes | |
| | Artificial surfaces | |
| Barely inflammable | Deciduous forest | 20 |
| Slightly inflammable | Mixed forest | 30 |
| Inflammable | Heathland | 50 |
| | Grassland | |
| | Fen | |
| | Reed land | |
| | Swamps | |
| Highly inflammable | Mixed heathland and forest | 60 |
| Extremely inflammable | Closed coniferous forest | 100 |





**Table 2.** Relative areas (%) per risk class for the three IPMs and the average probability assigned to the ignition points.

| Risk | Interval ($\times 10^{-6}$) | Land cover class | Land cover class & soil type | Land cover class, soil type & land use class |
|---|---|---|---|---|
| Low | 0.000 – 0.025 | 71.72 | 73.81 | 71.29 |
| Intermediate | 0.025 – 0.120 | 23.05 | 15.14 | 21.86 |
| High | 0.120 – 0.400 | 4.52 | 10.27 | 6.36 |
| Very High | > 0.400 | 0.71 | 0.78 | 0.48 |
| Score ($\times 10^{-6}$) | | 0.28 | 0.31 | 0.45 |



**Table 3.** The relative area (%) per risk class for the Belgian provinces and the capital region of Brussels.

| Region | Province | Low | Intm. | High | Very high |
|---|---|---|---|---|---|
| Flanders | Antwerp | 71.38 | 7.98 | 19.30 | 1.34 |
| | Flemish Brabant | 82.03 | 13.01 | 4.83 | 0.13 |
| | West-Flanders | 94.62 | 2.23 | 3.10 | 0.06 |
| | East-Flanders | 88.30 | 6.38 | 5.17 | 0.15 |
| | Limburg | 66.95 | 8.38 | 20.79 | 3.88 |
| Wallonia | Hainaut | 81.20 | 16.14 | 2.60 | 0.06 |
| | Walloon Brabant | 85.85 | 10.49 | 3.62 | 0.05 |
| | Liège | 63.42 | 33.69 | 2.89 | 0.00 |
| | Luxembourg | 46.20 | 49.26 | 4.39 | 0.15 |
| | Namur | 60.88 | 38.68 | 0.45 | 0.00 |
| | Brussels | 70.98 | 29.01 | 0.01 | 0.00 |





**Figure captions**

**Figure 1:** Land cover class (a), soil type (b), and land use class (c) for Belgium.

**Figure 2:** The expected and observed ignition frequency in relation to the distribution of the land cover classes (a), the soil classes (b), and the land use classes (c).

**Figure 3:** Wildfire ignitions in Belgium between 1994–2016.

**Figure 4:** the relative frequency of 376 ignitions between 1994–2016.

**Figure 5:** Number of ignitions per year.

**Figure 6:** The Belgian wildfire hazard map.

**Figure 7:** The average ignition probability observed in the data points that were not used for the construction of the IPM.

**Figure 8:** An illustration of the dependency on the number of data points of the robustness of the risk map. Part (a) shows the robustness of the risk map in function of the data period that was used for construction, from 1994 to the upper limit. Part (b) shows the actual number of data points, used for construction, from 1994 to the upper limit.

**Figure 9:** Frequency of the calculated probabilities in the IPMs constructed with land cover class (a), land cover class and soil type (b), and land cover class, soil type, and land use class (c) and the indication of the four risk class intervals.

**Figure 10:** The ignition probability map constructed with land cover class, soil type, and land use class.

**Figure 11:** Average calculated ignition probability per hazard class.




# Figures

**Figure 1.**

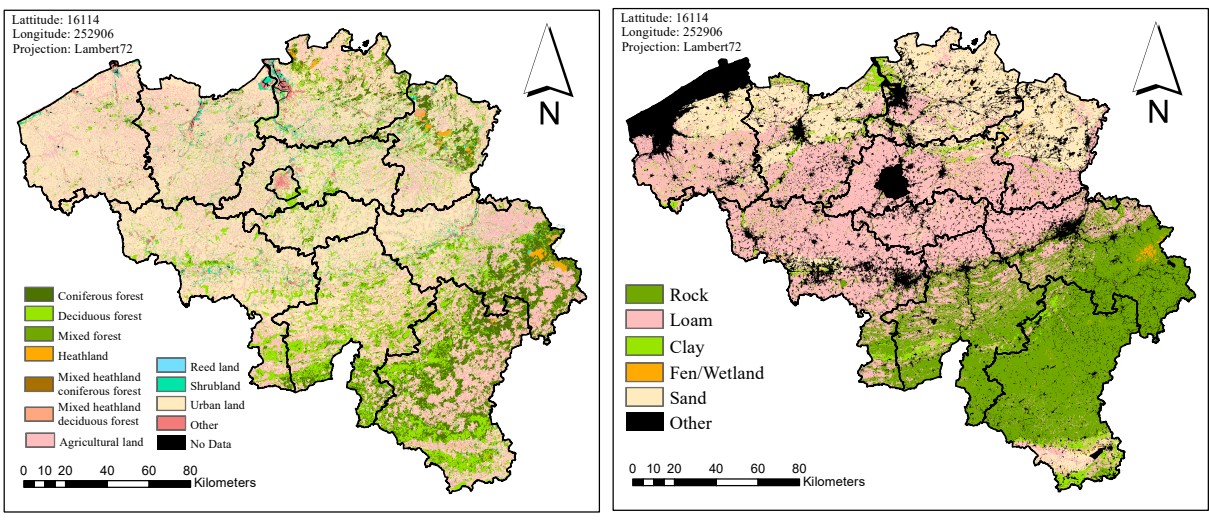

(a) Land cover class       (b) Soil type

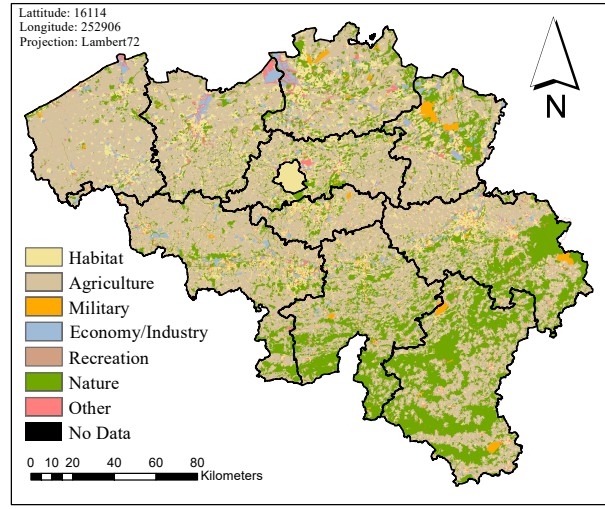

(c) Land use class





**Figure 2.**

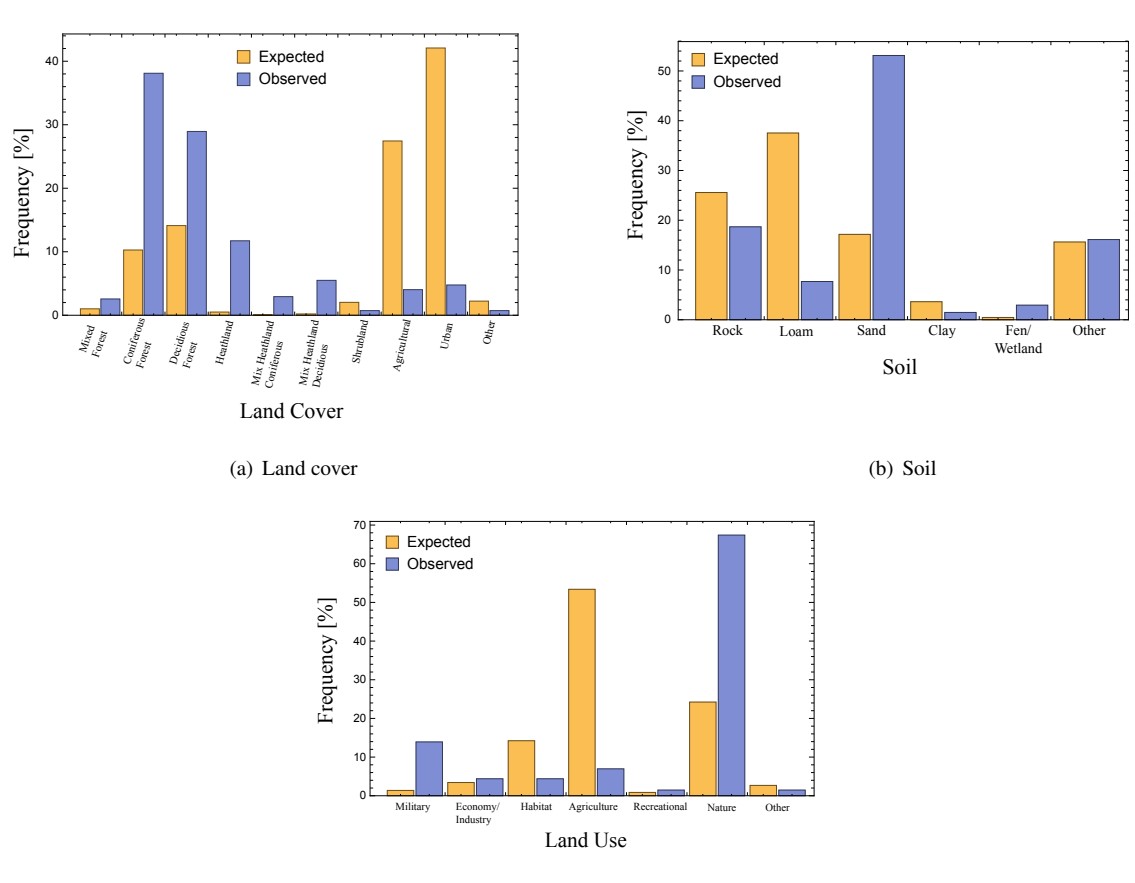

(a) Land cover

(b) Soil

(c) Land use





**Figure 3.**

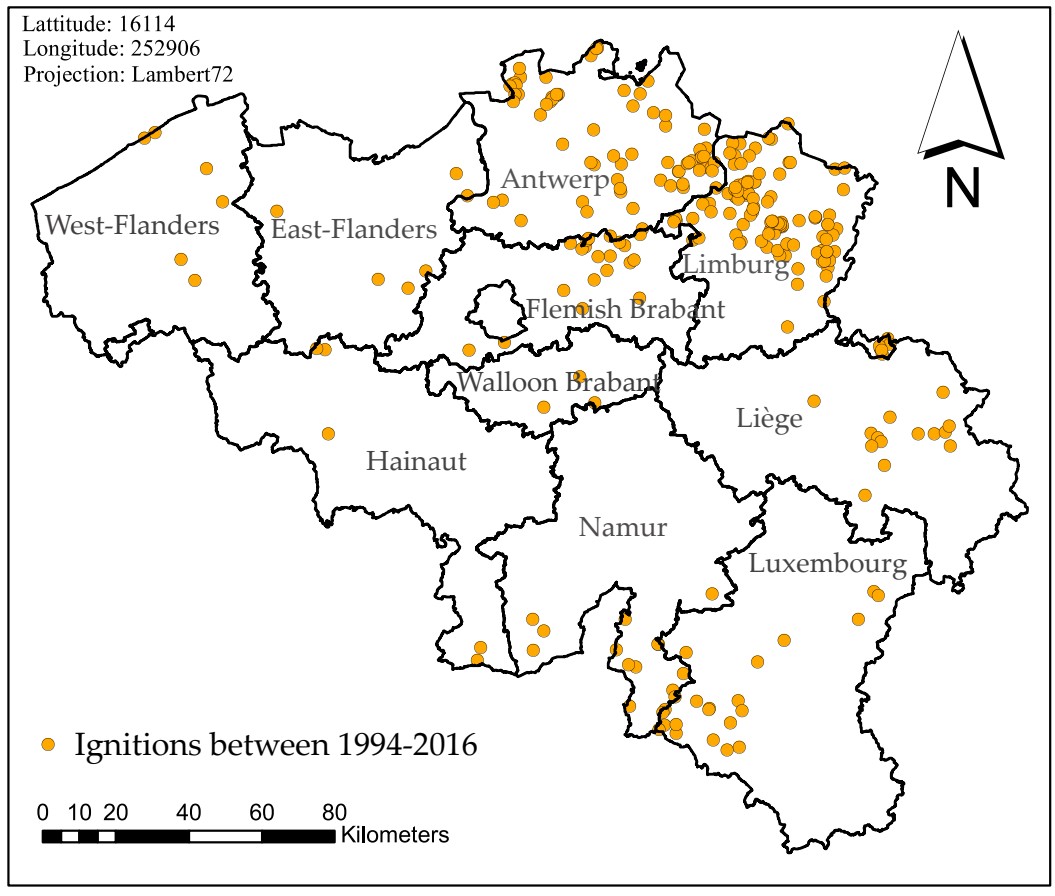





**Figure 4.**

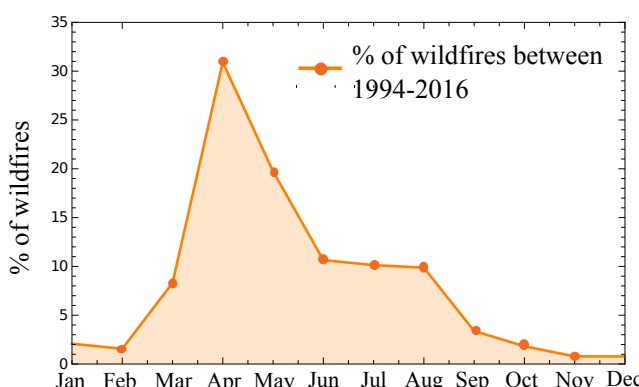





**Figure 5.**

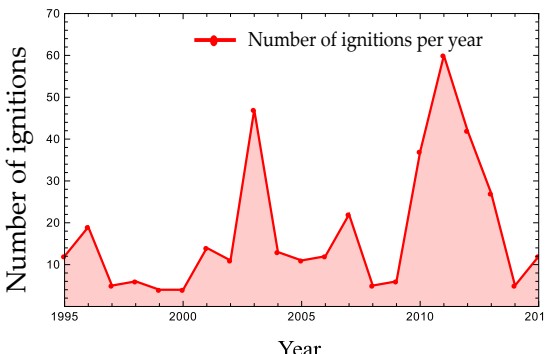





**Figure 6.**

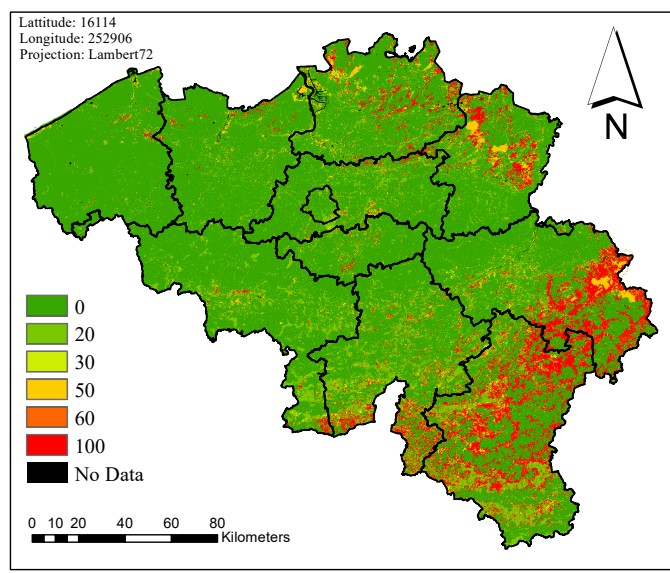





**Figure 7.**

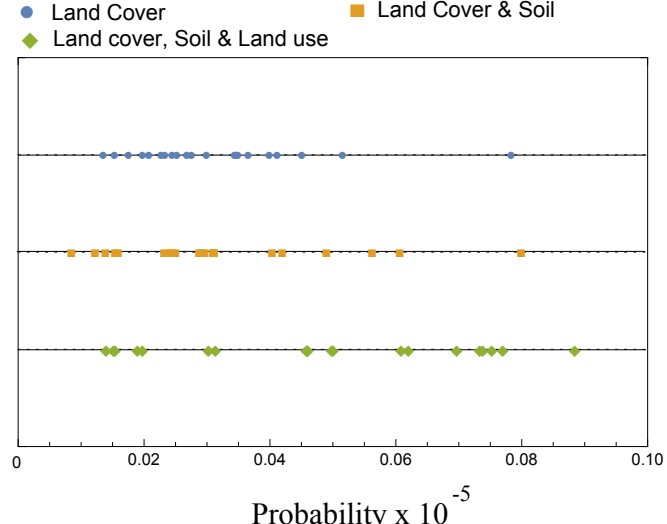





**Figure 8.**

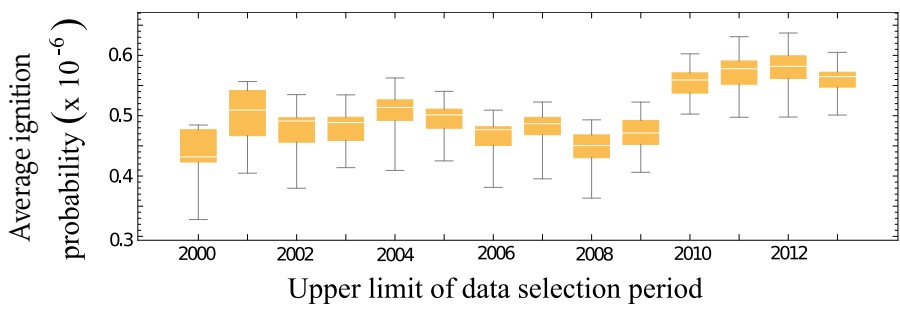

(a)

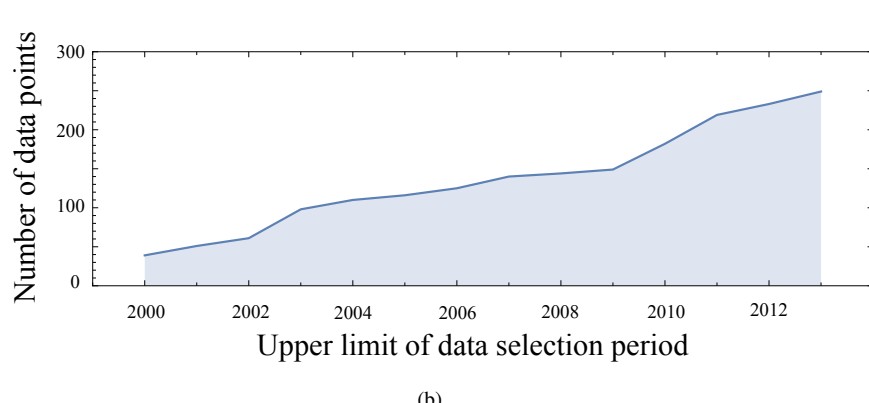

(b)

**Figure 9.**

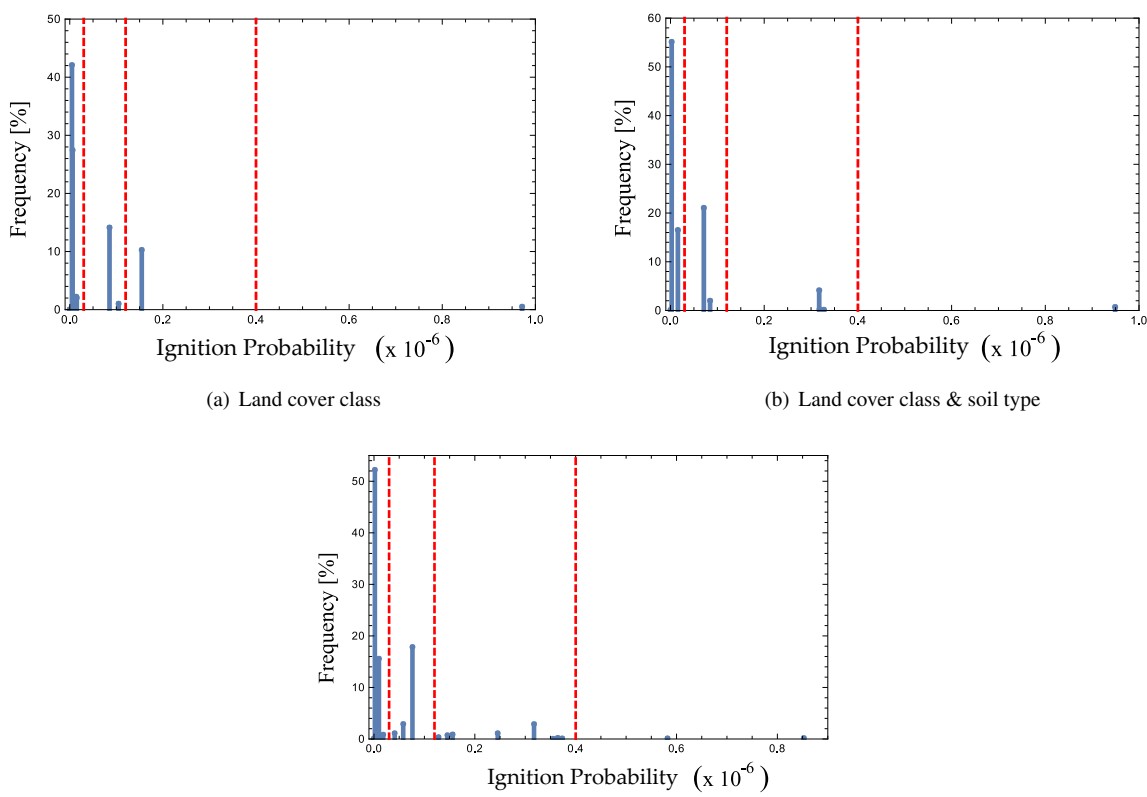

(a) Land cover class

(b) Land cover class & soil type

(c) Land cover class, soil type & land use class





**Figure 10.**







**Figure 11.**

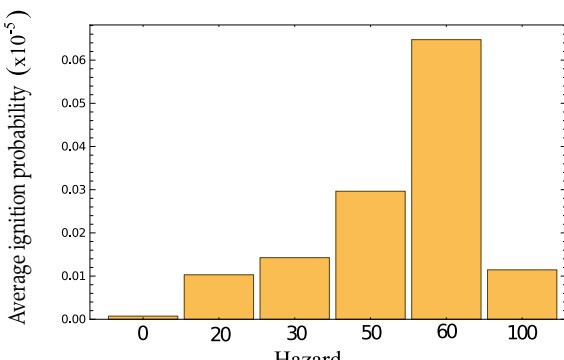