# Peer review of "Wildfire ignition probability in Belgium"

_Natural Hazards and Earth System Sciences, 2018_

## Referee Comment (RC1) · Anonymous Referee #1 · 28 Oct 2018

The manuscript presents a methodology for mapping wildfire ignition probability for Belgium using GIS overlay and expert rules techniques. It also provides a study of the fire regime in the country, based on historical data and analysis. The methodological approach, which is presented in the manuscript, is interesting and well described; adequate background information is given, which provides a comprehensive overview of the aspects that relate to wildland fires in Belgium. Below are some comments and suggestions for the revision of the paper (and the changing of its title). I believe that these are quite important in making the manuscript suitable for publication in the NHESS Journal. 1. The terms "Risk" and "Hazard" are not used as expected in a contemporary research work. The definition of "Risk" that is provided on page 4, line 23, was used in the past, but is nowadays avoided, since many terms related to natural hazards have been standardized. Therefore, the use of the term "risk assessment" in the title of the paper will be rather misleading for the readers of a

recent paper. It is essential that a new work contributes to the effort for establishing a common language for the study of natural hazards and specifically wildland fires, and complies with the existing EU directives and standards. Thus, I would suggest the revision of the manuscript, in order to use other, more appropriate terms for the description of the work and the maps produced. The INSPIRE document, which can be found via the following link, is an important source of information: https://inspire.ec.europa.eu/documents/Data_Specifications/INSPIRE_DataSpecification_NZ_v3.0.pdf (see p.126) 2. The "Hazard map" that has been produced in the context of this study is a reclassification of the land cover map, based on the "inflammability score" attributed to each land cover type. A name that includes the term "inflammability" or "combustibility" would be a more appropriate description of the content of this map. The following links are available to the authors, so as to study the concepts of "hazard potential" and "risk" that are used in the US: https://www.firelab.org/project/wildfire-hazard-potential https://www.firelab.org/sites/default/files/images/downloads/wfp_methods_041813.pdf 3. The inflammability categories and scores presented in table 1 are too generic for the creation of a map with reliable information for comparison with the "ignition probability map". Moreover, the scores seem quite strange for some land cover categories and especially for "closed coniferous forest". As the authors state on page 9, lines 23-24: "It seems that wildfires in such land cover are less controllable than those in coniferous or deciduous forests. This can be understood by the fact that heathlands and fens are largely covered by shrubs and grass that ignite easily". However, the table 1 scores that are used in the study are not consistent with this statement. A clarification is needed as to what are the species and forest structure expressed by the category "closed coniferous forest". Is it fir or pine, dense and closed high forests or other type of conifers? Verboom et al. (2013), in section 6.1, refers to "bushy conifer forest, juniper and rhododendron" and the score 100 may be reasonable for these, but not for fir or pine closed forests (eg. dense overstory but without understory vegetation). I would suggest using more criteria for the reclassification of the land cover map into a map that can express inflammability. This might also provide better results

in the comparison between the "probability of ignition" and "hazard" maps. Some additional comments below: 4. Page 6, lines 15-16: "The soil and land cover can serve as a proxy to fire susceptibility as soil texture is correlated with soil moisture. . ." Please justify your argument about the correlation of soil type, soil texture and soil moisture, and their relation to fire susceptibility. 5. Page 4, lines 15-20: Correctness and consistency are necessary in the description of the term "fire hazard". It is explained as "potential fire behavior" in line 15 and that it "expresses the potential of wildfire occurrence" in line 18. Although wildland fires are not currently considered a significant natural hazard in Belgium, both in terms of occurrence and in terms of impact, this paper, if revised appropriately, will become an interesting and informative work for the community that deals with wildfires, since climate change is expected to play a role in increasing the significance of fires in the whole of Europe in the future, especially in the wildland-urban interface environments.

---

## Referee Comment (RC2) · Anonymous Referee #2 · 6 Nov 2018

The manuscript "A first wildfire risk assessment for Belgium" presents a wildfire ignition probability map for Belgium using GIS techniques and probability rules. The database and applied methodology attempt to indicate the spatio-temporal manner of wildfire ignition in Belgium. The topic is within the scope of the Journal; wildfire is an environmental problem, which can cause a host of natural hazard and human health impacts. The manuscript focuses on an interesting topic, however I think there are some critical issues which should be considered before publishing the paper. The manuscript can be accepted after paying close attention to the following points and making the necessary revisions.

1.The authors are recommended to revise the structure of the paper and use of ă numbered ă outlines.

2.Anthropogenic factors and natural factors are known as the reasons behind the wild-

fire ignition in Belgium. It could be useful to provide the spatio-temporal map of these factors to characterize the superiority of human factors.

3.Page 6, lines 12-13 "The Flemish soil and land use layer date from 2016 and 2014, respectively, while those from Wallonia date from 2007 and 2016, respectively". The expressions "from 2016 and 2014" and "from 2007 and 2016" are not true. Please revise these expressions.

4.Page 2, line 18, Page 6, line 19, line 27, Page 7, line 10, line 16. What do you mean by "Section"? Please mention the name of the desired section.

5.Please use a flowchart in the methodology part of your manuscript to describe your methodology step by step.

6.Page 13, lines 4-6, "All together, it seems that the best way of preventing wildfires is perhaps to exclude human activities in the most fire prone areas and increase the awareness among the general public, so that people become more aware of the danger they pose to their natural environment". The proposed solution "to exclude human activities in the most fire prone areas" does not seem to be scientific and logical to prevent wildfire. Please search about other solutions and methods to prevent wildfire and control it.

7.According to the Figure 4, Although the summer is warmer than spring but most wildfires have occurred in the spring. Is there any scientific and specific reason behind the wildfire in spring?

8.As can be seen from Figure 5, the number of ignitions has increased between 2009 and 2015. Is there any relation between climate change and the number of ignitions in this period? I mean what is the reason behind the increasing the number of ignitions in this period?

9.The conclusion part is explained superficially. Please explain this part more precisely.

10.The authors are recommended to analyze climatic data (temperature and precipitation) in meteorological stations close to the wildfire places to understand the relation between climate condition and wildfire ignition. The authors can use the data of some sample places in the Belgium to clarify this topic.

I hope that these comments will help improve the manuscript.

Good luck.

---

## Author Comment (AC1) · 26 Dec 2018

Dear Madam/Sir,

Thank you for your revision of the manuscript 'A first wildfire risk assessment for Belgium'. In the attachment, you can find two pdf files. The first one is entitled 'Response to referee 1.pdf', and contains three chapters: (i) the referee's comments, (ii) the author's responses, and (iii) the revised manuscript, with all changes, additions, and deletions highlighted. The second file is a pdf file of the manuscript in its current form and is entitled 'Wildfire ignition probability in Belgium.pdf'.

I'd like to thank you once more for your constructive feedback, and I hope I was able to integrate the comments in the manuscript to your satisfaction.

Yours sincerely,

[Figure]

Arthur Depicker

Please also note the supplement to this comment:
https://www.nat-hazards-earth-syst-sci-discuss.net/nhess-2018-252/nhess-2018-252-AC1-supplement.zip

---

## Author Comment (AC3) · 9 Apr 2019

Dear sir/madam,

As we submitted the revision of this manuscript at the end of December 2018, I was wondering whether we may soon expect a decision.

Best wishes,

Jan Baetens
* * *

---

## Author Response (AR1)

**Referee 1**

**C1**

**The terms "Risk" and "Hazard" are not used as expected in a contemporary research work. The definition of "Risk" that is provided on page 4, line 23, was used in the past, but is nowadays avoided, since many terms related to natural hazards have been standardized. Therefore, the use of the term "risk assessment" in the title of the paper will be rather misleading for the readers of a recent paper. It is essential that a new work contributes to the effort for establishing a common language for the study of natural hazards and specifically wildland fires, and complies with the existing EU directives and standards. Thus, I would suggest the revision of the manuscript, in order to use other, more appropriate terms for the description of the work and the maps produced. The INSPIRE document, which can be found via the following link, is an important source of information: https://inspire.ec.europa.eu/documents/Data_Specifications/INSPIRE_DataSpecification_NZ_v3.0.pdf (see p.126)**

According to contemporary definitions, risk is the potential for realization of unwanted, adverse consequences to human life, health, property or the environment (Miller and Ager, 2013). According to existing EU directives, risk is the product of two components: fire hazard and vulnerability. Fire hazard is governed by (i) ignition and (ii) propagation. This study focuses on the spatial distribution of the ignitions, hence the title was changed to 'A first wildfire ignition probability assessment for Belgium'. The definitions in the text were adjusted, first explaining the Risk framework (p.3, 69-77) and then focusing on the 'ignition' part (p.3, 78-79).

**C2**

**The "Hazard map" that has been produced in the context of this study is a reclassification of the land cover map, based on the "inflammability score" attributed to each land cover type. A name that includes the term "inflammability" or "combustibility" would be a more appropriate description of the content of this map. The following links are available to the authors, so as to study the concepts of "hazard potential" and "risk" that are used in the US:**

**https://www.firelab.org/project/wildfire-hazard-potential
https://www.firelab.org/sites/default/files/images/downloads/wfp_methods_041813.pdf**

In line with the new Risk framework, the term 'hazard' is indeed misused throughout the paper, and I follow the recommendation of the reviewer to refer to the land cover map as the 'inflammability score'. Following this new terminology, I do not longer see the relevance of showing this map and decided to no longer include the inflammability map. In the current risk framework, the map merely reflects the land cover classes (Figure 2c), as opposed to the old framework in which such a map was an important component of the risk (being the hazard).

**C3**

**The inflammability categories and scores presented in table 1 are too generic for the creation of a map with reliable information for comparison with the "ignition probability map". Moreover, the scores seem quite strange for some land cover categories and especially for "closed coniferous forest". As the authors state on page 9, lines 23-24: "It seems that wildfires in such land cover are less controllable than those in coniferous or deciduous forests. This can be understood by the fact that**

heathlands and fens are largely covered by shrubs and grass that ignite easily". However, the table 1 scores that are used in the study are not consistent with this statement. A clarification is needed as to what are the species and forest structure expressed by the category "closed coniferous forest". Is it fir or pine, dense and closed high forests or other type of conifers? Verboom et al. (2013), in section 6.1, refers to "bushy conifer forest, juniper and rhododendron" and the score 100 may be reasonable for these, but not for fir or pine closed forests (eg. dense overstory but without understory vegetation). I would suggest using more criteria for the reclassification of the land cover map into a map that can express inflammability. This might also provide better results in the comparison between the "probability of ignition" and "hazard" maps.

Although I do not give a lot of attention to the inflammability map anymore, I can give a few notes on the original file, of which the original classes are displayed in **Table 1**. Although I most certainly agree that the tree species and forest density influence the flammability, there is not enough information in this data layer to extract this information. Moreover, the method I use does not allow very complex covariates (p.7, 228-233), especially given the size of the used wildfire database. This lack of detail in the land cover map has also lead to the decision not to include the inflammability map in the manuscript, because I would not be able to follow your recommendations.

**Referee 2**

**C1**

**The authors are recommended to revise the structure of the paper and use of a numbered outline.**

**C2**

**Anthropogenic factors and natural factors are known as the reasons behind the wildfire ignition in Belgium. It could be useful to provide the spatio-temporal map of these factors to characterize the superiority of human factors.**

On Figure 1, you can find a map of Belgium. On this map, I added information on the population density and the (major) military training areas. Two observations can be made here: (i) most wildfires occur in regions with low or intermediate population densities, and (ii) many ignitions occurred in the vicinity of military domains. The latter makes sense, as many wildfires are ignited through military exercises, but the former observation can be somewhat contraintuitive, given the fact that an inverse relationship is often observed (higher density -> more fires) (e.g. Catry 2009). Indeed, the situation in Belgium is different, as relatively high population densities can be found practically everywhere. One could argue that the presence of more people would imply more social control and hence less arson or a faster suppression and thus avoiding wildfires.

**C3**

**Page 6, lines 12-13 "The Flemish soil and land use layer date from 2016 and 2014, respectively, while those from Wallonia date from 2007 and 2016, respectively". The expressions "from 2016 and 2014" and "from 2007 and 2016" are not true. Please revise these expressions.**

**C4**

**Page 2, line 18, Page 6, line 19, line 27, Page 7, line 10, line 16. What do you mean by "Section"? Please mention the name of the desired section.**

**C5**

**Please use a flowchart in the methodology part of your manuscript to describe your methodology step by step.**

Flowchart is added as Figure 4

**C6**

**Page 13, lines 4-6, "All together, it seems that the best way of preventing wildfires is perhaps to exclude human activities in the most fire prone areas and increase the awareness among the general public, so that people become more aware of the danger they pose to their natural environment". The proposed solution "to exclude human activities in the most fire prone areas" does not seem to be scientific and logical to prevent wildfire. Please search about other solutions and methods to prevent wildfire and control it.**

(p.13, 499-506) I agree that this recommendation was somewhat radical and certainly not socially acceptable. I revised the text and concluded with 5 recommendations (i) excluding military activity in fire-prone areas during the fire season, (ii) improving collaboration with foreign emergency services, (iii) concentrating the dedicated resources in the areas that display the highest ignition probabilities (iv) improving fire detection methods, and (v) raising more awareness amongst the population.

**C7**

**According to the Figure 4, Although the summer is warmer than spring but most wildfires have occurred in the spring. Is there any scientific and specific reason behind the wildfire in spring?**

(p.10, 365-371) An important controlling factor for anthropogenic fires is drought (Burk, 2005), and it happens so that April is the driest month in Belgium (Journée et al., 2015), hence the peak in this month. I considered more detailed information (on why this is the driest month) out of the scope of this article.

**C8**

**As can be seen from Figure 5, the number of ignitions has increased between 2009 and 2015. Is there any relation between climate change and the number of ignitions in this period? I mean what is the reason behind the increasing the number of ignitions in this period?**

The main reason for this is the availability of a detailed database in the period 2010-2013 (and partly 2014) giving rise to an elevated number of ignitions. There is indeed no consistent registration of ignitions, making it difficult, if not impossible, to quantify the impact of climate (change) on ignition frequency.

**C9**

**The conclusion part is explained superficially. Please explain this part more precisely.**

I restructured the paper: I merged the results and discussion session and extended it in many parts (mostly guided by your previous comments) and wrote a new conclusion section. Furthermore, I wrote a new introduction section and reorganized the methods and results section in a more logical and chronological way.

**The authors are recommended to analyze climatic data (temperature and precipitation) in meteorological stations close to the wildfire places to understand the relation between climate condition and wildfire ignition. The authors can use the data of some sample places in the Belgium to clarify this topic.**

There are two reasons why I did not include climatological data: **(i)** the model can only deal with a limited amount of covariates (p.7, 228-233), and **(ii)** it is a challenging endeavor to find appropriate climatic variables (p.8, 271-285) I elaborated on the use of precipitation data (both the annual rainfall and the drought sensitivity). A quick glance at the two data layers (Figure 3) shows that neither variables are suitable for modelling. The wettest areas are known to be fire prone (Compare figure 3a and Figure 1) and the driest areas are almost free of wildfire occurrences. I am sure that drought has a strong impact on wildfires, but a more accurate proxy has to be found.

**Appendix**

Table 1: The land cover classes and the way they were reclassified

| | Original | Simplified |
|---|---|---|
| 1 | Coniferous | Coniferous forest |
| 2 | Mixed coniferous-deciduous with dominance of coniferous forest | Coniferous forest |
| 3 | Mixed coniferous-deciduous with no dominance | Mixed forest |
| 4 | Mixed coniferous-deciduous with dominance of deciduous forest | Deciduous forest |
| 5 | Deciduous forest (F) | Deciduous forest |
| 6 | Poplar  (Pp) | Deciduous forest |
| 7 | Tree nurseries (Os) | Agricultural |
| 8 | Orchards (V) | Agricultural |
| 9 | Bushes (B) | Shrubland |
| 10 | Heathland (La) | Heathland |
| 11 | Heathland with some coniferous trees (Las) | Mixed heathland Coniferous Forest |
| 12 | Heathland with some deciduous trees (Las) (Laf) | Mixed heathland Deciduous Forest |
| 13 | Mixed heathland and shrubs (Lab) | Heathland |
| 14 | Non-specified growth of annual plants/herbs (K) | Shrubland |
| 15 | Non-specified growth of annual plants/herbs with some trees and shrubs (Kb) | Shrubland |
| 16 | Reedland (Ro) | Reedland |

| | | |
|---|---|---|
| 17 | Permanent pasture and hayland (P) | Agricultural |
| 18 | Grasspatch (G) | Urban |
| 19 | Garden (J) | Urban |
| 20 | Built-up land (C) | Urban |
| 21 | Bare sand (Z) | Other |
| 22 | Bare rock (Rx) | Other |
| 23 | Graveyard (Cx) | Other |
| 24 | Not specified (X) | Other |

---

## Author Response (AR2)

**Response to Reviewing committee comments: Wildfire ignition probability in Belgium**

**Arthur Depicker**

September 23, 2019

We would like to thank the referees for their detailed and constructive comments. We believe that their feedback identified some weaknesses in our methodology and discussion. Through completing the suggested edits, the revised manuscript benefits substantially from an improvement in the results, overall presentation, and clarity.

More specifically, thanks to the useful comments of the reviewers, we refined our explanations of the introduced concepts in the paper and we updated our methods by changing the spatial resolution at which the analyses were performed (100 m instead of 10). In general, these alterations in the models' set-up did not result in major changes in model outcomes and consequent interpretation. Other comments and suggestions made by the referees are discussed point by point below. To elaborate our answers to the reviewers' comments, the following color scheme is used: comments of the referees are shown in **blue**, answers are in black and quotes from the revised text are in **green**. The lines in the final manuscript are indicated in **purple**, while the lines in the manuscript with tracked changes are in **orange**.

**Contents**

| 1        | Referee #1         1.1 Minor comments | 1
1 |
|----------|---------------------------------------|---------------|
| 2 | Referee #2                            | 2      |
|          | 2.1 Major comments                    | 2
9        |

**1 Referee #1**

**1.1** Minor comments**

**1.1** Congratulations to the authors for the improvement of the paper and the change of the title. I think that it is an interesting work. I suggest to to the authors to do some more changes, for consistency purposes and replace the term "risk" where it is used in the results, conclusions and tables with a more appropriate description of the concepts "risk class", "risk map" e.t.c. (i.e Ignition Probability).

The reviewer's concern is justified: we look, indeed, at the ignition probability, and not the risk. We substituted the term 'wildfire risk' with 'wildfire ignition probability' where relevant (earlier on in the methods section we do talk about risk):

L360/L372, L370/L382, L478/L505, L480/L507, L524/L554, Table 2/Table 2, L751-753/L787-788, L756/L791, L758/L794

**2 Referee #2**

This is an interesting topic, within the scope of the journal, and especially appealing given that, as stated by the authors, the number of ignitions in Belgium are likely to increase due to climate change. The paper is clearly written and data and methods are clearly described. However, from my point of view, there is a certain number of aspects that put at stake the results presented and the conclusions reached in the paper. This is why I recommend that this paper should not be accepted at this stage, although I encourage the authors to look at my criticism (that I intended to be constructive) and resubmit it at a later stage.

**2.1 Major comments**

**2.1** My first major concern is the choice of a 10m spatial scale for this study (1st paragraph of section 2.4). Why did the authors chose such a small scale? And how did they handle the implications of such choice? Here are some of my concerns:

**2.1.1** Given that the majority of ignitions were extracted from newspapers, how did the authors assign the location at a 10m scale? and, for those events where there is GPS data, is the precision less than 10m? and even so, is it really required to locate ignitions at a 10m scale?

We share the concern of the reviewer. For the wildfire data, we only have a description of the ignition location, or, if applicable, a residential address. A resolution of 10 m would indeed be too optimistic, so we adopted a resolution of 100 m for all covariates. We justified this choice in the methods section: - for the government data:

**L201-203**/**L204-206**: The ignition location was identified by means of (i) a residential address, (ii) personal communication with the fire-fighting services, and/or (iii) topographic features.

- for the newspaper data:

**L209-211/L212-214**: For these instances, the location of the wildfire ignition was assessed using (i) the description of topographic features, and (ii) communications with the relevant fire-fighting services.

Given the uncertainties tied up with the ignition locations, we can then speculate about the valid spatial resolution for the data:

**L211-213/L214-216**: This way, we assumed that the remaining uncertainty on the location of the registered wildfires was higher than the chosen 100 m spatial resolution.

Due to the use of a lower spatial resolution, the results changed slightly. As to the comparison of the three IPMs, the same patterns were observed, namely that the third IPM has by far the highest average ignition probability among the observed wildfires. Figure 7 in the manuscripts was updated accordingly:

As was the text in the manuscript:

**L458-461/L476-480**: The Mann-Whitney U test showed that there was no significant difference in the medians of  $IPM_1$ , and  $IPM_2$  (p = 0.561). However, the  $IPM_3$  had a significantly higher median than  $IPM_1$  (p = 0.020) and  $IPM_2$  (p = 0.003). Hence,  $IPM_3$ , based on three covariates, was considered the best wildfire ignition probability model.

The analyses of the robustness was updated as well (Fig. 1):

**Methods:**

L344-354/L353-366: The first map was constructed with data from the period 1994–2004. Subsequently, we incrementally increased the length of the period from which data were used in the IPM construction stage with one year. As such, we constructed 13 IPMs, the first one with data from the period 1994–2004, the last one with data from the period 1994–2004, the last one with data for the period 1994–2016. For each IPM, we randomly selected 90% of the data for calibration, while the remaining 10% of the instances was used to assess the quality, i.e. the average predicted probability within observed ignition points. The robustness of each of the 13 IPMs was tested by calibrating each of the IPMs 100 times. This approach allowed us to construct a boxplot of the corresponding average ignition probabilities in the 13 IPMs. The range of each of these 13 probabilities is a proxy for the robustness of the IPMs.

Results:

**L462-467/L481-487**: From Fig. X, we infer that the quality of the IPM, expressed as the predicted probability in observed ignition points, remains stable for an increasing inventory. It can also be observed that the robustness of the IPM increases substantially for the smaller datasets, while, for datasets larger than the one that contains the data from the period 1994–2011 (219 ignitions), the quantiles of the boxplots appear at more or less the same values.

The new final probability map is almost a copy of the one in the original manuscript (Fig. 2).

Figure 1: The new figure for the robustness/quality assessment

---

## Author Response (AR3)

**Response to Reviewing committee comments:**
**Wildfire ignition probability in Belgium**

**Arthur Depicker**

**November 7, 2019**

We would like to thank the referees for their constructive comments. The minor revisions have identified some lacunae in the conclusions of the manuscript. According to these latest suggestions of the reviewers,the manuscript improved in terms of clarity and nuance in the drawn conclusions.

More specifically, we extended our conclusion, including some remarks on (i) the impact of (i) the spatial resolution of our model, and (ii) the omission of weather conditions. To elaborate our answers to the reviewers' comments, the following color scheme is used: comments of the referees are shown in **blue**, answers are in black and quotes from the revised text are in **green**. The lines in the final manuscript are indicated in **purple**, while the lines in the manuscript with tracked changes are in **orange**.

**1    Referee #3: Minor revision**

*I recognize the effort made by the authors to answer to the points I raised and I think that the manuscript has improved substantially.*

- *I am still not convinced by the spatial scale chosen. Is 100 m still compatible with the type of available information about fire events? And is this resolution really required for the type of maps the authors intend to generate? I realize that this is not an easy question but, as pointed by the authors, when degrading from 10 m to 100 m had impacts on results (e.g. on Figure 7).*

- *I am also not convinced by the choice made by the authors to neglect weather conditions. However I realize that it would be difficult to incorporate this information at the current stage of the manuscript.*

*I therefore strongly suggest that these two issues (and their possible implications) are addressed by the authors in their concluding remarks.*

As to the first remark of the reviewer: we're confident that the lowering of the spatial resolution has not led to major differences in the probability map (Figure 1). The resolution had also no impact on our conclusions drawn with regard to the most optimal model. To better justify the scale we used (and why it has to be so low), we adapted the first paragraph of the conclusions:

**L498-503**/**L498-503**   *The study was complicated by (i) the lack of literature on wildfires in Belgium, (ii) the limited number of ignitions,* **and (iii) the uncertainty of the ignition locations. The latter was a decisive factor in determining the optimal spatial resolution of the model, i.e. low enough to capture the uncertainty on the ignition**

*data while high enough to allow for the application of our model at a provincial or municipal scale.*

To answer the second remark: we do expect that meteorological conditions are an important factor for wildfire ignitions. Given that we want to produce a static probability map, we are restricted to 'aggregated' meteorological (= climatic) covariates. We briefly discussed 'precipitation' and 'drought sensitivity', yet came to the conclusion that these are not optimal for ignition prediction on an annual scale. We better clarified our meaning in the methods section:

**L318-321**/**318-321** *Given the fact that most anthropogenic wildfires are controlled by drought (Burk et al., 2005), we advise future research to develop more suitable proxy variables for drought in Belgium **that reflect the different responses of different plant communities and soil types to precipitation deficits.***

and we stressed the potential added value of better proxy variables in the conclusion:

**L520-534**/**L520-534** *In order to calculate the ignition probability, we used a straightforward data-driven approach relying on Bayes' rule. Contrary to other approaches (X et al., XXXX), the resulting map provides a tangible estimation of the annual probability that a wildfire will ignite in a given pixel of 100 by 100 m. Moreover, we demonstrated that this approach can be used to obtain an estimate of the average annual ignition probability in a certain area. Our method involved the delineation of environments through the combination of predictor classes. Because of the limited number of wildfires, it was necessary to limit the number of environments to 20, and hence the number of covariates to three. **To allow for more covariates, an expansion of the ignition database would be necessary.** It could be concluded that the approach relying on exactly three covariates (land cover, soil, and land use) led to the most reliable wildfire ignition probability map, which is, moreover, robust to an increase in the number of wildfires in the underlying database. **We assume that our model could be substantially improved through the inclusion of more covariates, preferably a drought index for Belgium that reflects plant moisture sensitivity to precipitation deficits.***

[Figure]

Figure 1: a) probability map on 10 m resolution and b) on 100 m resolution.